# Hydrophobic Polymeric Additives toward a Long-Term Robust Carbonaceous Mudstone Slope

**DOI:** 10.3390/polym13050802

**Published:** 2021-03-05

**Authors:** Hongyuan Fu, Caiying Chen, Huanyi Zha, Du Yuan, Qian-Feng Gao, Ling Zeng, Chuankun Jia

**Affiliations:** 1School of Traffic & Transportation Engineering, Changsha University of Science & Technology, Changsha 410114, China; chency100_18@163.com (C.C.); qianfeng.gao@csust.edu.cn (Q.-F.G.); 2School of Civil Engineering, Changsha University of Science & Technology, Changsha 410114, China; zhahy0326@stu.csust.edu.cn (H.Z.); zl001@csust.edu.cn (L.Z.); 3College of Materials Science and Engineering, Changsha University of Science & Technology, Changsha 410114, China; aduyuandu@gmail.com

**Keywords:** carbonaceous mudstone, hydrophobic material, mechanical properties

## Abstract

Slopes with carbonaceous mudstone (CM) are widely distributed in the southwest of China and have experienced numerous geological disasters in special climate, especially in rainfall conditions. Therefore, novel materials to stabilize CM slopes have attracted increasing interests. However, developing ultra-stable and cost-effective additives for CM slopes is still a great challenge. Herein, a hydrophobic polymeric material (polyvinylidene fluoride, PVDF) is investigated as an additive to enhance the mechanical strength and long-time stability of CM slopes. The PVDF is uniformly dispersed in CM matrix via interfacial interaction. The contact angle of the PVDF-modified carbonaceous mudstone (PVDF-MCM) can reach as high as 103.95°, indicating an excellent hydrophobicity. The unconfined compressive strength (UCS) and tensile strength (TS) of PVDF-MCM have been intensively enhanced to 4.07 MPa and 1.96 MPa, respectively, compared with ~0 MPa of pristine CM. Moreover, the UCS and TS of PVDF-MCM remain at 3.24 MPa and 1.03 MPa even after curing for 28 days in high humidity conditions. Our findings show that the PVDF can improve the hydrophobicity of CM significantly, which leads to super mechanical stability of PVDF-MCM. The excellent performance makes PVDF a promising additive for the development of ultra-stable, long-lifetime and cost-effective carbonaceous mudstone slopes.

## 1. Introduction

Uneven shrinkage, water-weakening effects, temperature effects, and mechanical changes are the main factors contributing to the damage of microstructures of rock blocks [1]. Water plays a key role in mudstone disintegration [2]. The interaction between water and mudstone causes significant changes in physical and mechanical properties of the rock mass. Mechanical properties including the compressive strength of mudstone decrease with the extension of immersion time and gradually stabilize [3]. Carbonaceous mudstone (CM) is widely distributed in the southwest of China, where the annual precipitation in this region is typically 1200–2000 mm (e.g., in the year 2015) or even more [4]. CM shows disintegration when undergoing wetting–drying cycles and temperature change. After disintegration, its strength sharply drops with large wetting deformation and secondary disintegration continues. As a result, layered disintegration happens in CM from the outside towards the inside. Consequently, the disintegration results in massive waste of solid CMs. Due to the above, CM slopes easily show instability and collapse [5,6]. During the “Twelfth Five-Year Plan” period, 64,521 geological disasters (including collapses, landslides, mudslides, ground collapse, ground fissures, ground subsidence, etc.) occurred in China, resulting in 2008 deaths and missing people, 1317 injuries, and direct economic losses of 27.34 billion RMB [7]. Therefore, research on the stability of mudstone slopes is critical and imperative.

Common methods for enhancing the stability of slopes include slope protection, anchors, anti-slide piles and comprehensive reinforcement methods [8,9,10,11], but none of them are based on improving the engineering performance of slope itself. Although traditional calcium-based rock and soil amendments (e.g., cement, lime, and fly ash) can effectively improve the strength and durability of slope, they may produce cracks and reduce the service life of slope [12,13,14]. Especially for the easy-to-disintegrate CM slope under the wetting–drying cycles, traditional calcium-based materials cannot effectively solve the layered disintegration phenomenon. For this reason, non-calcium-based materials including polymers and emulsions have been used as soil stabilizers to increase strengths, reduce soil liquefaction and permeability, and improve water and weathering resistances. For example, superabsorbent polymers have been used to improve the moisture sensitivity and the shear strength of subgrade soil [15]. Emulsions have been receiving increasing attention due to their good stability, low cost, and easy workability. Some emulsions such as methylene diphenyl diisocyanate, xanthan gum, styrene–acrylic emulsion, amphiphilic O/W emulsions, water-based polyurethane, and bitumen emulsions are successfully used as additives to optimize the mechanical strength and other engineering properties of rock or sand slopes [16,17,18,19,20,21,22]. However, the above materials are mainly used to modify soils, where their applications on disintegrated CM are rarely reported. Polyvinylidene fluoride (PVDF) is a typical hydrophobic material, which is commonly used in water treatment, membrane distillation, gas separation, lithium-ion battery separators, composite membrane, and other fields, owing to its high thermal stability, good chemical properties, and strong film forming properties [23,24].

In this work, the PVDF is proposed to modify carbonaceous mudstone. The unconfined compressive strength (UCS) and tensile strength (TS) (including durability) of the PVDF modified carbonaceous mudstone (PVDF-MCM) are investigated, with the characterizations of scanning electron microscopy (SEM), X-ray fluorescence (XRF), X-ray diffraction (XRD), and Fourier transform infrared spectroscopy (FTIR) tests. The relationship between the contact angle and strength of modified CM and the PVDF contents are evaluated qualitatively and quantitatively. Further, the interaction between PVDF and CM is discussed from the perspectives of microstructure and chemical analysis, where the interaction mechanism is proposed.

## 2. Experimental Details

### 2.1. Materials

CM was taken from the K18 + 500 site of the Liuzhai–Hechi Expressway in Guangxi. Previous tests showed that after disintegration under a certain vertical load and wet–dry cycles, the proportion of CM particles is smaller than 2 mm [5]. So, CM particles were dried at 106 °C for 48 h, then cooled to room temperature and passed through a 2 mm sieve. The particle-size distribution of CM is shown in Figure 1. The mineral composition of CM included quartz, kaolinite, mica, siderite, pyrite, etc., with mass fractions of 40.28, 24.86, 18.79, 8.52, 4.65, and 2.9 wt%, respectively. The maximum dry density, optimal moisture content, liquid limit, and plastic limit of the main physical parameters of CM were 2.08 g/cm^3^, 10.78, 33.10, and 25.20 wt%, respectively.

PVDF has the molecular formula of -(CH2-CF2)_n_-, where strong hydrogen bond exists in the closely arranged molecular chains. The SEM image in Figure 2a shows that PVDF particle size is about 200 nm and uniformly distributed. Figure 2b shows the FTIR spectrum of pure PVDF, 873, 1067, 1181, and 1402 cm^−1^ peaks are assigned to the vibrational features of C–C skeleton, C–F, stretching of –CF_3_ and deformation of –CH_2_, respectively [25,26].

### 2.2. Methods for Preparing Samples

PVDF was fully dissolved with organic solvent to form a PVDF solution. A certain amount of dried CM was weighed and uniformly mixed with the prepared PVDF solution to obtain a mixture. The PVDF content was defined as the mass ratio of PVDF to CM, and its values were 4, 8, 10, and 12 wt%; PVDF content of 0 wt% was also considered for comparison purposes (denoted as M-4, M-8, M-10, M-12, and M-0, respectively). In this study, the compaction degree, dry density, and the organic solvent content of each sample were 96%, 2.01 g/cm^3^, and 20%, respectively. The sample preparation process and reinforcement ideas are shown in Figure 3.

### 2.3. Analytical Methods

#### 2.3.1. Contact Angle Tests

Contact angles of PVDF-MCM with different PVDF contents were measured by the mercury intrusion method. A certain amount of mixture was weighed, and the sample was prepared by static pressure method, which had a diameter of 61.8 mm and a height of 20 mm. The tests were conducted on the Mercury Porosimeter (SDC-350 contact angle measuring instrument). An initial droplet volume of 1 μL and a photo frequency of 150 frames per second were applied in the tests. The contact angle tests were repeated three times for each sample, and the average values were taken for subsequent analyses.

#### 2.3.2. Mechanical Tests

Unconfined compressive strength (UCS) test is one of the engineering properties of embankment fillers. UCS test is performed to study the effect of PVDF on the compressive strength of modified CM (MCM). The UCS samples were prepared following these procedures: (i) A certain mass of mixture was poured into a steel mold in three layers, and each layer was compacted using the static compaction method to ensure the integrity of the sample, and at the same time, the layers were lapped. (ii) After compaction, both ends of the sample were flattened, then the mold was removed to form a sample with a diameter of 50 mm and a height of 100 mm. (iii) Because the melting point of PVDF is about 160 °C [26,27], the samples were dried for 6 h at 140 °C and for 2 h at 160 °C, then cooled down to the room temperature to achieve the best molding effect. (iv) The samples were placed in a standard room (the temperature is 20 ± 2 °C and the relative humidity is ≥95%) for curing. The curing periods were 0, 3, 7, 14, 28 days for durability test. UCS test was conducted in accordance with the Chinese standard “Test methods of soils for highway engineering” [28]. The test instrument was an electronic universal testing machine, and the tests were conducted at an axial strain rate of 1%/min.

TS can directly reflect the mutual attraction between soil particles and the cohesion of agglomerate materials [29]. The production of tensile sample is to press a certain amount of mixture at one time, and then get it through the same drying steps as UCS and measure it by the direct test method. According to Nahlawi et al. [30], Li et al. [31], and Huang et al. [32], the tensile testing device was improved, and an “8” type loading fixture was designed. The tensile testing device and sample sizes are shown in Figure 4. The TS durability test is the same as the UCS curing conditions and periods.

#### 2.3.3. Microscopic and Compositional Tests

XRD, FTIR, SEM, and XRF tests were carried out on PVDF-MCM samples of M-0 and M-10, both of which were at a curing age of 0 d. In SEM tests, typical test blocks with fracture surface about 1 cm^2^ of unconfined compressive samples were taken [33].

## 3. Results and Discussion

### 3.1. Contact Angle

The time-dependent contact angle is analyzed in Figure 5. Figure 5a presents the optical images of M-0 after dropping 2 mL pure water on the surface for 0, 5, and 10 min, where the water droplets quickly penetrate deep into the sample within 10 min. Figure 5b further shows when the PVDF content is 0 wt%, the droplet almost penetrates into the sample at 1500 ms; while, when PVDF content is 10 wt%, the contact angle of droplet is stable at 103.95° at 5 min. Figure 5c shows images of M-10 after dropping 2 mL pure water on the surface for 0, 30, and 60 min, where the modified sample still retain its hydrophobicity after 60 min.

The contact angles of the modified CM with different PVDF contents in the initial state and stable state are shown in Figure 5d. It can be seen that with the increase of PVDF content, the initial contact angles increase gradually, reaching 98.23° when the PVDF content is 4 wt%. The stable contact angle of M-4 is almost unchanged compared to M-0. However, the stable contact angle increases rapidly with the increase of PVDF content, reaching 103.95° when the PVDF content is 10 wt%, where the corresponding initial contact angle is 123.85°. When the contact angle is larger than 90°, it means that the modified CM changes from hydrophilic to hydrophobic material [34]. The time-dependent contact angle is further analyzed.

Polymeric coating techniques are widely applied for tuning the surface properties, e.g., styrene-acrylic emulsion-modified weak rock presents a contact angle of 80° [19]. Our result shows that the PVDF modification on CM has successfully made the composite CM into hydrophobic material with good retention of hydrophobicity.

### 3.2. Mechanical Strength

The measured mechanical strength is shown in Figure 6. Figure 6a shows the UCS results of PVDF-MCM. UCS increases with the increase of PVDF content, reaching a maximum of 4.07 MPa when the PVDF content is 8 wt%, and then shows a decrease of 5% to 3.88 MPa when the PVDF content is 10 wt%. This shows that PVDF has an adverse effect on UCS of CM when its content exceeds 8 wt%. Figure 6b shows the TS results of PVDF-MCM. The M-0 samples break and lose their integrity once they are slightly touched, and the TS is considered to be 0 MPa. With the increase of the PVDF content, the TS increases significantly. When the PVDF content is 10 wt%, the TS is 1.47 MPa, and when the PVDF content is 12 wt%, the TS is 1.96 MPa. When the PVDF content does not exceed 12%, PVDF has a positive effect on the TS of the CM.

The relation between the durability and contact angle of M-10 is presented in Figure 6c,d. Figure 6c shows that in the early and middle stages of curing (i.e., 0, 3, 7 d), the UCS value is reduced by about 5% in turn, and in the later stage of curing (i.e., 14 and 28 d), the UCS is stable at 3.24 MPa, with 84% of the initial value retained. Figure 6d shows that in the early and middle stages of curing (i.e., 0, 3, and 7 d), TS is reduced by about 10% in turn, in the later stage of curing (i.e., 14 and 28 d), TS is stabilized at 1.03 MPa, with 70% of the initial value retained. These indicate that the high humidity at the pre-mid stage of curing have certain adverse effects on the UCS and TS of PVDF-MCM. Nevertheless, the UCS is stabilized after 14 d and is still 21.6 times higher than M-0, and the TS is infinite times higher than M-0.

The optimal UCS values of several modified soils are listed in Figure 6e [35,36,37,38,39,40], where the UCS of PVDF-MCM in our work is very prominent. On the other hand, the TS values of many modified soils does not exceed 0.5 MPa under the corresponding optimal conditions [32,41,42,43,44,45]; However, in our work, the TS of PVDF-MCM can reach ~2 MPa (Figure 6f) [46]. The results show that the PVDF-MCM has good mechanical properties, and the influence of humidity on the strength of PVDF-MCM can be efficiently alleviated.

### 3.3. Microstructure and Composition

Figure 7 shows the microstructure of M-0 (a and b) and M-10 (c and d) before curing [47]. It can be seen from Figure 7a,b that the CM particles have obvious boundaries and smooth and flat surfaces, but with a large number of large pores. Comparing Figure 7a,b with Figure 7c,d, it is clear that the particles are interlaced and overlapped, and their boundaries become blurred. PVDF-MCM particles have binder on the surface and between the particles, the pores of the CM are filled, and the whole structure is denser [48]. The pore size distribution (PSD) curve in Figure 8a obtained by the mercury intrusion porosimetry (MIP) [47] shows the pore diameter of samples are concentrated in the range of 1–3 μm. With the increase of PVDF content, CM particles bond more closely, which makes the pore diameter corresponding to the peak value of PSD curve show a gradually decreasing trend. The XRF results of M-0 and M-10 shown in Table 1 further confirm the binder composition, where the content of F in the modified CM increased significantly with 10 wt% PVDF as designed.

XRD analysis was carried out on PVDF, M-0, and M-10 to determine the phase of PVDF-MCM. The results are shown in Figure 8b. The diffraction angles 2θ of 18.42°, 20.00°, 26.66°, and 38.64°, are assigned to the characteristic α phase of PVDF [49]. The mineral compositions of M-0 are mainly quartz, pyrite, kaolinite, siderite, mica, calcite, hemihydrate gypsum. For M-10, it consists of the above mineral phases and the introduced PVDF phase. Hence, there is no formation of new minerals observed during the processing, where the quartz content is the maximum difference but within 1%. Typical diffraction peaks of mica and kaolinite are clearly revealed in M-10, comparing to M-0, the diffraction peaks of 8.92°, 12.43°, 17.87°, 24.97°, 27.93°, 45.54° show broadening, indicating the reduction in crystalline size of CM after processing. In M-10 there are also red-shifts of 0.06°, 0.04°, 0.08°, 0.06° for 8.92°, 17.87°, 27.93°, 45.54° peaks in mica phase and 0.06°, 0.06° for 12.43°, 24.97° in kaolinite phase suggest the increase of the interplanar spacings in CM particles, which is attributed to the change of interplanar spacings due to the introduction of PVDF in CM [49]. In order to further explore the interaction between PVDF and CM, the molecular structure and chemical properties of PVDF-MCM are then analyzed.

Figure 8c presents the FTIR results of pure PVDF, M-0, and M-10. It can be seen that M-0 and M-10 mainly contain a 3695 cm^−1^ internal surface hydroxyl vibration peak, which is located between the Al–O octahedron and Si–O tetrahedron, at 3621 cm^−1^ is an internal hydroxyl vibration peak and the Si–O vibration peak corresponding to 997 cm^−1^ [50,51]. Furthermore, three emerging absorption peaks in the spectrum of M-10, i.e., 1402 cm^−1^, 1181 cm^−1^, and 873 cm^−1^, correspond to the CH_2_, CF_3_, and C–C characteristic functional groups of PVDF. The consistent vibrational bands of M-10 with those of the pristine PVDF (Figure 2b) confirm the successful incorporation of the PVDF in CM. Interestingly, the broad vibrational feature at 3300–3500 cm^−1^ in M-0, due to the intercalated water in CM, is obviously reduced in M-10. This is consistent with the above finding on the change of interlayer distances between layer structures in both mica and kaolinite phases. Hence, by introducing PVDF into the CM system, the interlayer interaction in CM host is altered, which leads to the change of interlayer spacing.

## 4. Reinforcement Mechanism

PVDF solution is mixed with CM to form a hydrophobic structural film, and the loose CM is combined into a whole to effectively prevent the softening effect of water. The reinforcement mechanism of PVDF modifiers is as follows [19,52,53,54]:(i)Waterproof: After the evaporation of organic solvent, PVDF solidifies to form hydrophobic macromolecular polymer layer. It can be entangled with CM to form a network structure layer, which effectively prevents water from infiltrating into the material [55].(ii)Binding: Fluorinated chain segment shields the polarity of polar groups, which makes the surface of PVDF hydrophobic. At the same time, the surface energy of the PVDF membrane is low, which leads to the poor wettability of the membrane. PVDF has good cementation and can fill the pores of CM, which effectively increases the bond strength of CM particles and solidifies the particles into a new whole [56,57].(iii)Polymerization: PVDF and CM have good adhesion after mixing, because PVDF can be polymerized to form a polymeric layer. The oxygen-containing functional groups on the surface of CM particles may form chemical bonds with PVDF, which may enhance the mechanical strength of CM matrix (Figure 9). After modification, it has high strength and good hydrophobic properties.

## 5. Conclusions

In this work, a hydrophobic material, PVDF, was innovatively used to modify the mechanical strength and other properties of CM which was prone to continuous disintegration in water. The SEM images and FTIR results showed that PVDF dispersed uniformly in CM matrix, which might lead to a good strength of PVDF-MCM composites. The UCS and TS of PVDF-MCM were enhanced to 4.07 MPa and 1.96 MPa, respectively, both of which were much higher than those of pristine CM matrix. The experimental result of contact angle was much higher than 90°, implying an excellent hydrophobicity of PVDF-MCM composites. Importantly, the UCS and TS of PVDF-MCM are still high even in high humidity conditions. All results showed that PVDF-MCM had good hydrophobicity and mechanical properties. The modification mechanism was also analyzed and summarized. The PVDF-MCM mechanism revealed in this paper can provide some references for the application of new and non-traditional modifiers such as polymers, and the mechanism needs further verification by more experimental studies. The good performance and low cost of PVDF make it a promising candidate for stable CM slope application.

## Figures and Tables

**Figure 1 polymers-13-00802-f001:**
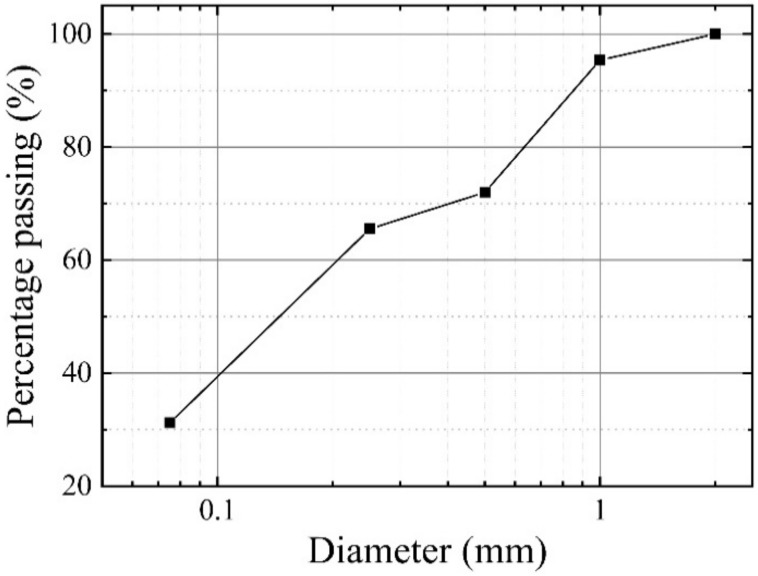
Size distribution of carbonaceous mudstone (CM) particles.

**Figure 2 polymers-13-00802-f002:**
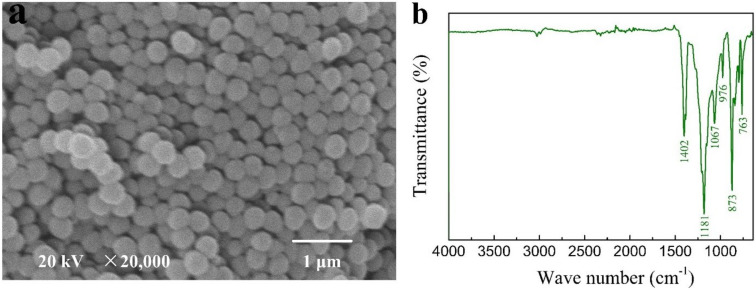
SEM image (**a**) and FTIR spectrum (**b**) of polyvinylidene fluoride (PVDF).

**Figure 3 polymers-13-00802-f003:**
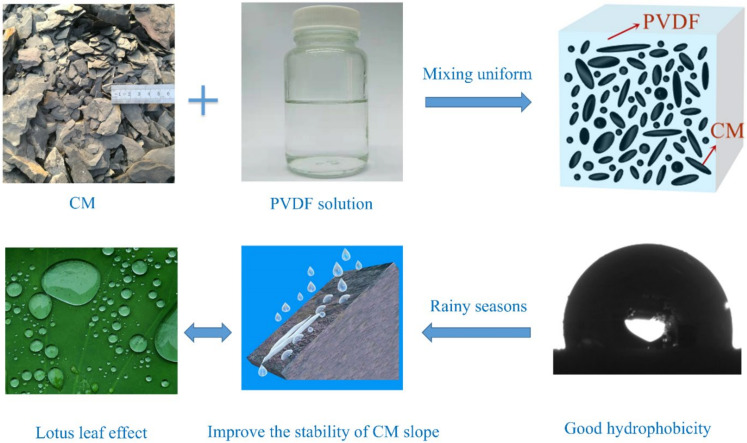
Schematic diagram of hydrophobic material (i.e., PVDF)-modified CM.

**Figure 4 polymers-13-00802-f004:**
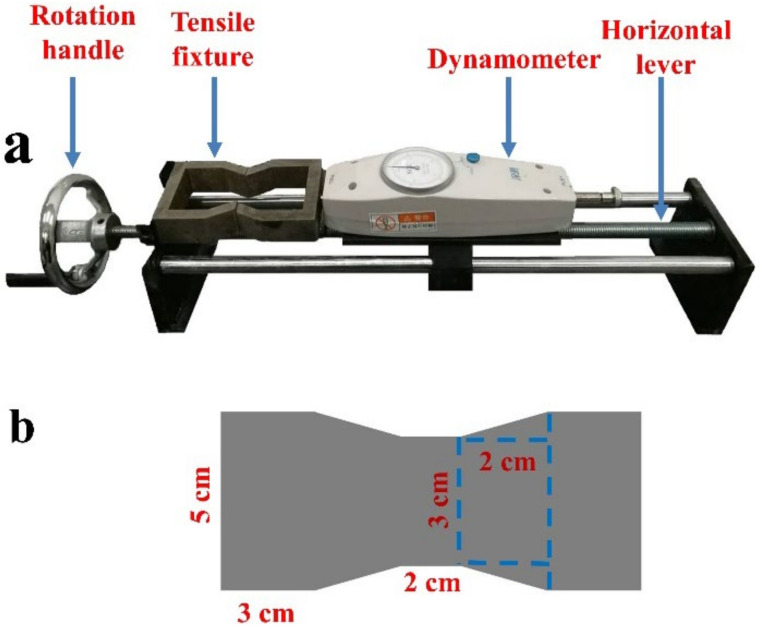
Device (**a**) and sample dimension (**b**) for tensile testing.

**Figure 5 polymers-13-00802-f005:**
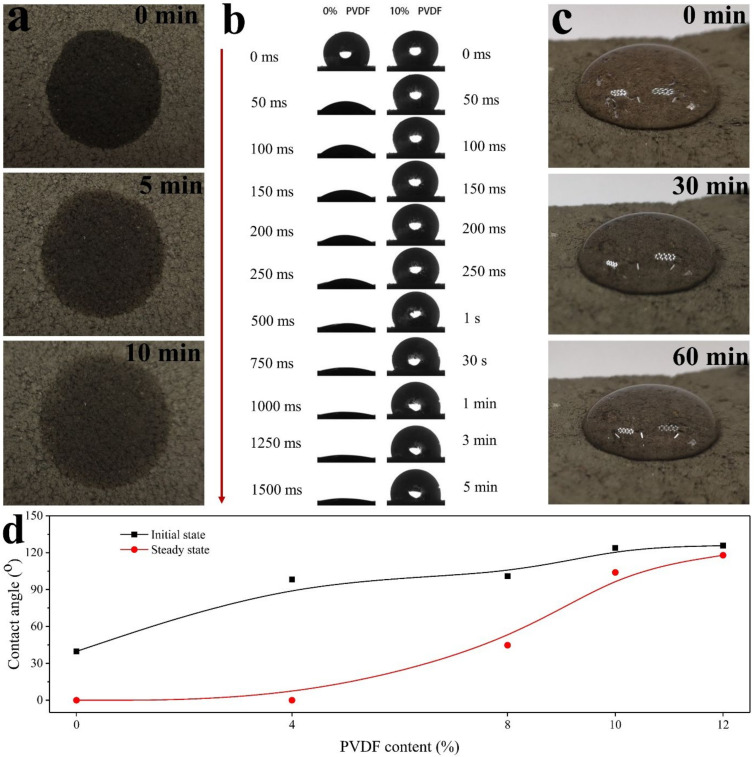
Wettability of PVDF-modified carbonaceous mudstone (PVDF-MCM). Water droplet (**a**,**c**) and contact angle (**b**) evolution on M-0 (**a**) and M-10 (**c**); (**d**) Contact angles of the modified CM with different PVDF contents.

**Figure 6 polymers-13-00802-f006:**
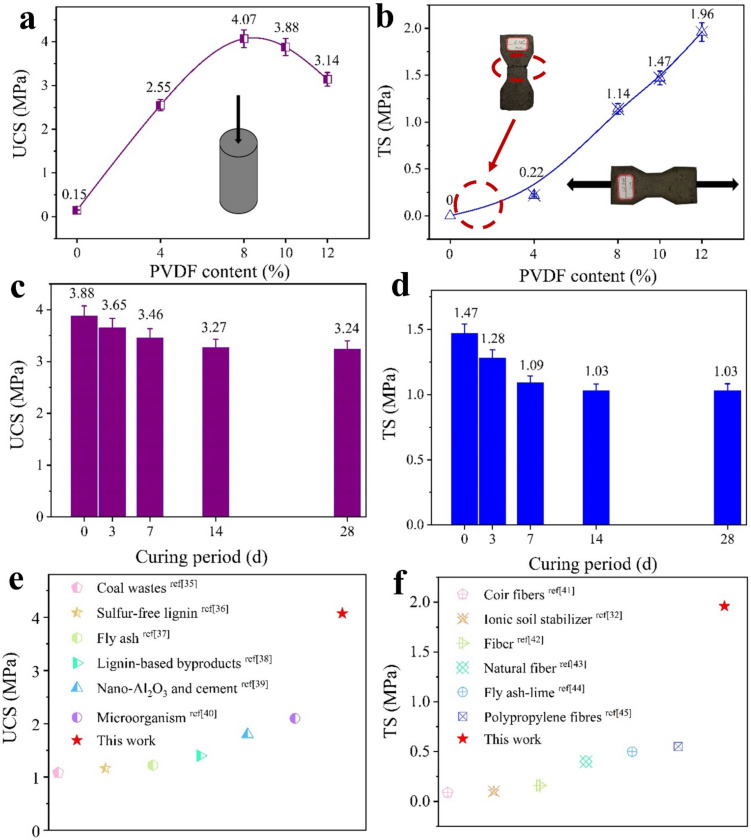
Mechanical properties of PVDF-MCM. Unconfined compressive strength (UCS) (**a**) and tensile strength (TS) (**b**) of PVDF-MCM with different contents; Influence of curing period on UCS (**c**) and TS (**d**) for M-10; Optimum UCS (**e**) and TS (**f**) for several reinforcement materials.

**Figure 7 polymers-13-00802-f007:**
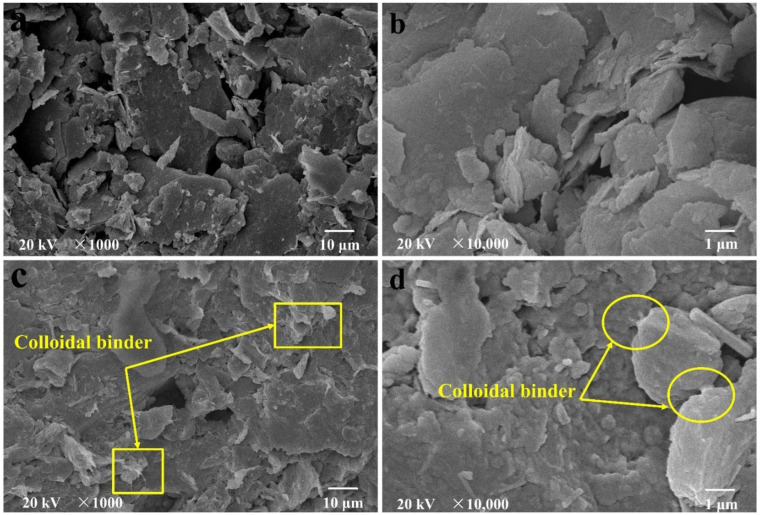
The microstructure of M-0 (**a**,**b**) and M-10 (**c**,**d**) before curing.

**Figure 8 polymers-13-00802-f008:**
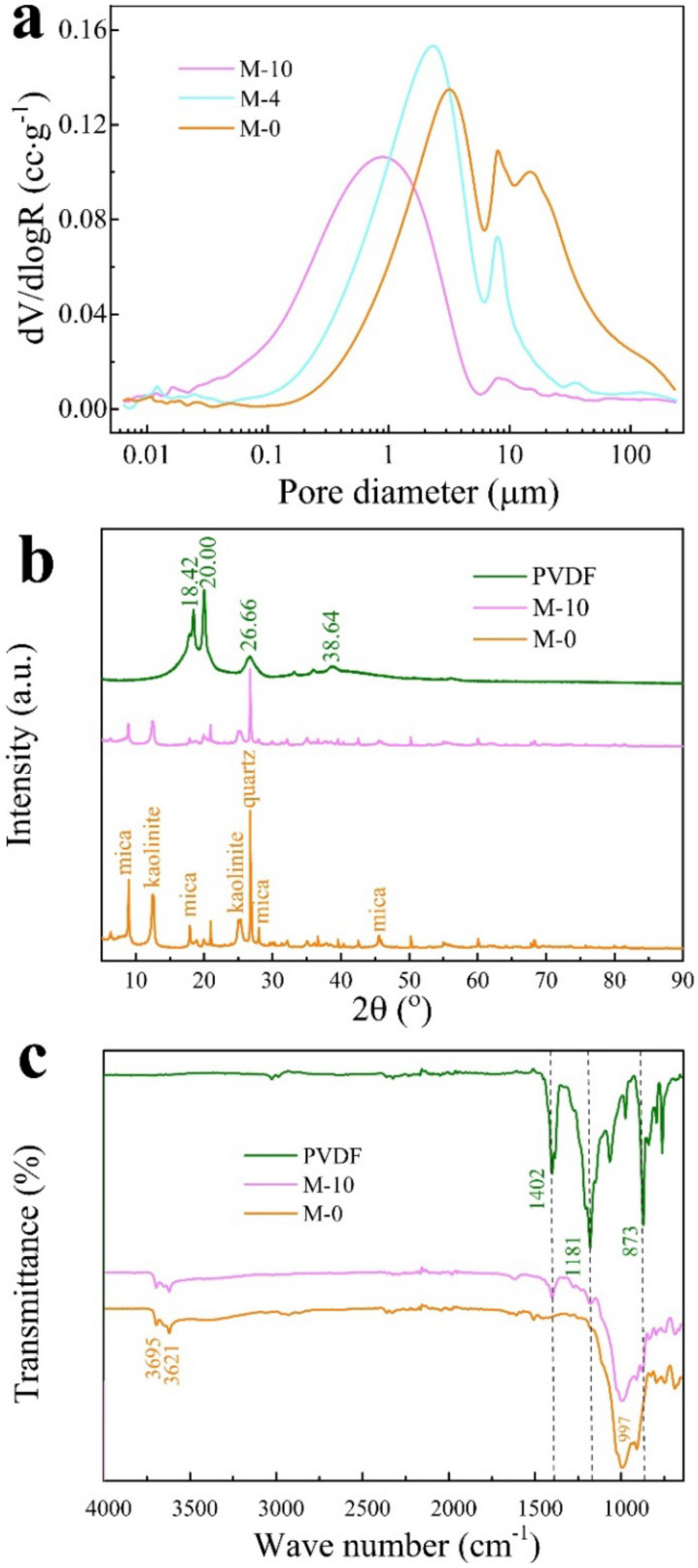
Pore size distribution (PSD) (**a**) of M-0, M-4 and M-10; XRD (**b**) and FTIR (**c**) of PVDF, M-0 and M-10.

**Figure 9 polymers-13-00802-f009:**
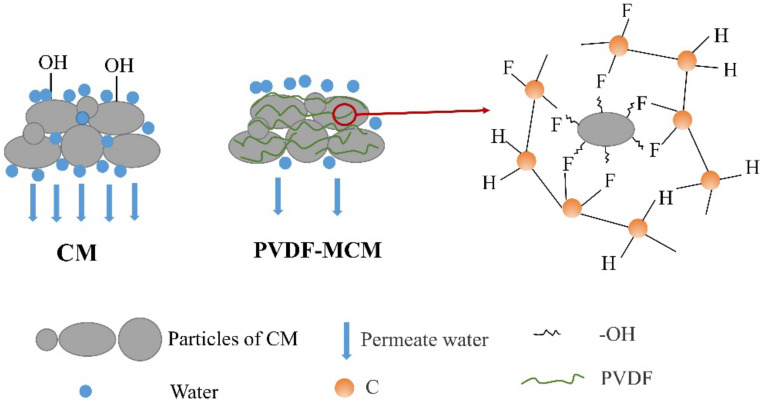
Schematic diagram of reforcement mechanism for PVDF-MCM.

**Table 1 polymers-13-00802-t001:** XRF of M-0 and M-10.

Samples	O	Si	Al	Fe	K	F	Ca	S	Mg	Others
M-0	43.40	22.48	13.98	4.17	3.31	0.00	1.76	0.79	0.64	9.47
M-10	40.30	21.39	13.21	4.55	3.08	2.94	1.76	0.78	0.62	11.37

## Data Availability

Both experimental and fitting data are available from the corresponding author.

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
