# Peer review of "Hydrophobic Polymeric Additives toward a Long-Term Robust Carbonaceous Mudstone Slope"

_polymers, 2021, doi:10.3390/polym13050802_

Round 1
Reviewer 1 Report
A paper with a minor review can be published. The conclusion needs to be connected more with the abstract. Is the contact angle related to any method? Is there a possibility to do another method of characterization?
Author Response
Dear Editor and Reviewers:
Thank you for giving us the opportunity to revise our manuscript. The comments are all valuable and very helpful for improving the quality of the manuscript. We have considered all comments carefully and have made relevant corrections in the revised manuscript. All comments have been addressed and the changed contents in the manuscript are marked in RED font. We hope the revised manuscript will meet your expectations and we are willing to answer any other questions you might have.
Herein, we offer detailed and point-by-point responses to the reviewers’ comments.
A paper with a minor review can be published. The conclusion needs to be connected more with the abstract. Is the contact angle related to any method? Is there a possibility to do another method of characterization?
Response: We sincerely appreciate the reviewer for the positive comments. The manuscript has been revised carefully.
Point 1: The conclusion needs to be connected more with the abstract.
Response 1: Thanks for your good comment. The conclusion had been revised accordingly and are shown as follows:
In this work, a hydrophobic material, PVDF, was innovatively used to modify the mechanical strength and other properties of CM which was prone to continuous dis-integration in water. The SEM images and FTIR results showed that PVDF dispersed uniformly in CM matrix, which might lead to a good strength of PVDF-MCM compo-sites. The UCS and TS of PVDF-MCM were enhanced to 4.07 MPa and 1.96MPa, respectively, both of which were much higher than those of pristine CM matrix. The experimental result of contact angle was much higher than 90o, implying an excellent hydrophobicity of PVDF-MCM composites. Importantly, the UCS and TS of PVDF-MCM are still high even in high humidity condition. All results showed that PVDF-MCM had good hydrophobicity and mechanical properties. The modification mechanism was also analyzed and summarized. The PVDF-MCM mechanism revealed in this paper can provide some references for the application of new and non-traditional modifiers such as polymers, and the mechanism needs further verification by more experimental studies. The good performances and low cost of PVDF makes it a promising candidate for stable CM slopes application.
Point 2: Is the contact angle related to any method? Is there a possibility to do another method of characterization?
Response 2: We are sorry for inaccurate definition of the method for measuring contact angle with “pendant drop method”. The method we used in this work is “mercury intrusion method”, which is most commonly used method for measuring contact angle. We have corrected the method as “mercury intrusion method” in the revised manuscript. The details of instrument are also provided in the revised manuscript.

Reviewer 2 Report
The manuscript "Hydrophobic Polymeric Additives on a Long-term Robust 2 Carbonaceous Mudstone Slope" described as author-said novel materials of modificated carbonaceous mudstone (CM).
In this work, a hydrophobic polymeric material (polyvinylidene fluoride, PVDF) was investigated as an additive to enhance the mechanical strength and long-time stability of CM slopes. The results are interesting but in my opinion, the minor corrections need. Authors should carefully consider the following suggestions before accepting for publication in Polymers.
- The introduction should be more specific
- The information about the modification process including Figure 1 should be in the section experimental data.
- The conclusion should be refined, because it is too lack.
- Reinforcement Mechanism is the most important in this material and should be enhanced. If it is possible, figure could be useful for understanding.
Author Response
Dear Editor and Reviewers:
Thank you for giving us the opportunity to revise our manuscript. The comments are all valuable and very helpful for improving the quality of the manuscript. We have considered all comments carefully and have made relevant corrections in the revised manuscript. All comments have been addressed and the changed contents in the manuscript are marked in RED font. We hope the revised manuscript will meet your expectations and we are willing to answer any other questions you might have.
Herein, we offer detailed and point-by-point responses to the reviewers’ comments.
In this work, a hydrophobic polymeric material (polyvinylidene fluoride, PVDF) was investigated as an additive to enhance the mechanical strength and long-time stability of CM slopes. The results are interesting but in my opinion, the minor corrections need. Authors should carefully consider the following suggestions before accepting for publication in Polymers.
Response: We appreciate the reviewer’s concerns on the various aspects of our paper. We believe our manuscript will be better articulated.
Point 1: The introduction should be more specific.
Response 1: We thank the reviewer for the valuable suggestion. We have revised the introduction in the revised manuscript. More additives have been introduced in the revised manuscript.
Point 2: The information about the modification process including Figure 1 should be in the section experimental data.
Response 2: The Figure 1 has been moved in the section of experimental data and it has been changed as figure 3 in the revised manuscript.
Point 3: The conclusion should be refined, because it is too lack.
Response 3: The conclusion had been revised in the revised manuscript and are shown as follows:
In this work, a hydrophobic material, PVDF, was innovatively used to modify the mechanical strength and other properties of CM which was prone to continuous dis-integration in water. The SEM images and FTIR results showed that PVDF dispersed uniformly in CM matrix, which might lead to a good strength of PVDF-MCM compo-sites. The UCS and TS of PVDF-MCM were enhanced to 4.07 MPa and 1.96MPa, respectively, both of which were much higher than those of pristine CM matrix. The experimental result of contact angle was much higher than 90o, implying an excellent hydrophobicity of PVDF-MCM composites. Importantly, the UCS and TS of PVDF-MCM are still high even in high humidity condition. All results showed that PVDF-MCM had good hydrophobicity and mechanical properties. The modification mechanism was also analyzed and summarized. The PVDF-MCM mechanism revealed in this paper can provide some references for the application of new and non-traditional modifiers such as polymers, and the mechanism needs further verification by more experimental studies. The good performances and low cost of PVDF makes it a promising candidate for stable CM slopes application.
Point 4: Reinforcement Mechanism is the most important in this material and should be enhanced. If it is possible, figure could be useful for understanding.
Response 4: Thanks for the helpful suggestion. To better understand the reinforcement mechanism of PVDF-MCM, a Schematic diagram of reforcement mechanism is shown as follows (Figure 9 in revised manuscript) [1-3]. The Oxygen-containing functional groups on the surface of CM particles may form chemical bonds with PVDF, which lead to an enhancement of the mechanical strength of CM matrix.
[1] J.L. Lv, G.Q. Zhang, H.M. Zhang, F.L. Yang. Graphene oxide-cellulose nanocrystal (GO-CNC) composite functionalized PVDF membrane with improved antifouling performance in MBR: behavior and mechanism. Chem. Eng. J. 352 (2018) 765-773.
[2] C.R. Wu, W.Y. Tang, J.H. Zhang, S.H. Liu, Z.Y. Wang, X. Wang, X.L. Lu. Preparation of super-hydrophobic PVDF membrane for MD purpose via hydroxyl induced crystallization-phase inversion. J. Membrane Sci. 543 (2017) 288-300.
[3] F. Jeschull, J. Maibach, K. Edstrom, D. Brandell. On the electrochemical properties and interphase composition of graphite: PVdF-HFP electrodes in dependence of binder content. J. Electrochem. Soc. 164 (7) (2017) 1765-1772.
